# A Case Report of Chronic Stress in Honey Bee Colonies Induced by Pathogens and Acaricide Residues

**DOI:** 10.3390/pathogens10080955

**Published:** 2021-07-29

**Authors:** Elena Alonso-Prados, Amelia-Virginia González-Porto, José Luis Bernal, José Bernal, Raquel Martín-Hernández, Mariano Higes

**Affiliations:** 1Unidad de Productos Fitosanitarios, Instituto Nacional de Investigación y Tecnología Agraria y Alimentaria (INIA, CSIC), 28040 Madrid, Spain; aprados@inia.es; 2Laboratorio de Mieles y Productos de las Colmenas Centro de Investigación Apícola y Agroambiental, IRIAF, Consejería de Agricultura de la Junta de Comunidades de Castilla-La Mancha, 19180 Marchamalo, Spain; avgonzalezp@jccm.es; 3Analytical Chemistry Group, Instituto Universitario Centro de Innovación en Química y Materiales Avanzados (I.U.CINQUIMA), Universidad de Valladolid, 47011 Valladolid, Spain; joseluis.bernal@uva.es (J.L.B.); jose.bernal@uva.es (J.B.); 4Instituto de Recursos Humanos para la Ciencia y la Tecnología (INCRECYT-FEDER), Fundación Parque Científico y Tecnológico de Castilla—La Mancha, 02006 Albacete, Spain; rmhernandez@jccm.es; 5Laboratorio de Patología Apícola, Centro de Investigación Apícola y Agroambiental, IRIAF, Consejería de Agricultura de la Junta de Comunidades de Castilla-La Mancha, 19180 Marchamalo, Spain

**Keywords:** honey bees, *Apis mellifera*, acaricides, pesticides, toxic unit, *Varroa destructor*, *Nosema ceranae*, bee viruses, tau-fluvalinate, coumaphos

## Abstract

In this case report, we analyze the possible causes of the poor health status of a professional *Apis mellifera iberiensis* apiary located in Gajanejos (Guadalajara, Spain). Several factors that potentially favor colony collapse were identified, including *Nosema ceranae* infection, alone or in combination with other factors (e.g., BQCV and DWV infection), and the accumulation of acaricides commonly used to control *Varroa destructor* in the beebread (coumaphos and tau-fluvalinate). Based on the levels of residues, the average toxic unit estimated for the apiary suggests a possible increase in vulnerability to infection by *N. ceranae* due to the presence of high levels of acaricides and the unusual climatic conditions of the year of the collapse event. These data highlight the importance of evaluating these factors in future monitoring programs, as well as the need to adopt adequate preventive measures as part of national and international welfare programs aimed at guaranteeing the health and fitness of bees.

## 1. Introduction

Bees, including honey bees, bumble bees, and solitary bees, are a prominent and economically important group of pollinators worldwide. In fact, 35% of the global food crop production depends on these pollinators [1], and in Europe, the production of 84% of crop species is, to some extent, dependent on animal pollination [2]. Bees also fulfill an important role in the pollination of wild plants. Thus, habitat fragmentation seems to have a negative effect on pollination and plant reproduction [3]. However, there is still a debate on how to approach the pollen limitation in plant dynamics [4,5]. Furthermore, honey bees provide additional economic inputs in temperate areas where honey production is a fundamental source of income to professional beekeepers.

The honey bee colony is a complex system in which thousands of individuals work together to ensure its sustainability. Multiple factors play an important role in colony viability, such as climate, environment, nutrition, and pathogens, and consequently, in colony pollination and production capabilities. Thus, any deterioration of honey bee colonies has a direct negative environmental impact and, in the case of honey bees, economic consequences in countries where there is a large proportion of professional beekeepers as in Mediterranean areas [6].

Many factors have been related to the decline in honey bee colonies over the past decades (revised in [7]). On one hand, the global spread of pathogens related to colony losses entails changes to the ecological and evolutionary dynamics of both pathogens and hosts, often leading to the selection of the most virulent variant of the pathogen and reducing the heterogeneity of the host. A range of treatments exist to combat pathogens and diseases during the last decades. Thus, chemicals that keep *Varroa* mite populations under control may accumulate in different hive matrices, chronically exposing honey bees to the residues of chemicals. Due to foraging activities, honey bees may also be exposed to a wider range of potentially toxic compounds (naturally produced or not), which may also accumulate inside the hive. The action of pathogens and xenobiotics or the combination of both provokes physiological changes, immunosuppression, and gut microbiota disruption on honey bees, which may finally produce the collapse of colonies. These changes may be also influenced by the nutrition quality of the collected pollen, which depends on the seasonal climatological conditions [8,9,10]. Overall, the impact of pathogens on this decline (mainly parasites and related viruses) may be particularly important [11,12,13], probably in conjunction with the accumulation of pesticide residues in hive matrices. Thus, the combined effect of two or more such stressors could drive the mortality of individuals, eventually leading to colony collapse [14,15,16].

Accordingly, we present here a screening study of a professional Spanish apiary that reported a problematic health situation. To investigate the factors that had possibly provoked this situation, we performed a comprehensive evaluation of multiple drivers, including pathogens and pesticides, also analyzing the foraging flora, in an attempt to determine whether new factors should be examined in future monitoring programs.

Severe *Nosema ceranae* infection, in conjunction with the accumulation of acaricides used to control *Varroa* mite infestations in honey bee hives, represents a real threat to colonies, indicating that appropriate preventative strategies should be adopted in honey bee health programs.

## 2. Results

### 2.1. Veterinary Inspection

A professional beekeeper who managed 400 honey bee colonies (*Apis mellifera iberiensis*) in Gajanejos (Guadalajara, Central Spain; lat. 40.8423, long. −2.8933) reported health problems in his colonies. He revised all his colonies in September 2015 while applying a compulsory treatment for *Varroa destructor* with CheckMite^®^ strips (a.m.: coumaphos), according to the manufacturer’s recommendations. No health problems had previously been noted in the colonies, which appeared to be in satisfactory health at the beginning of autumn, September 2015, with a normal population of worker honey bees. When the acaricide strips were removed 6 weeks after their application (November 2015), the beekeeper noticed a reduction in worker honeybee population in some of the colonies, and consequently, he began to inspect the apiary more often. The first dead colonies were detected that winter, and losses continued until the next spring, March 2016.

Upon veterinary inspection in early March 2016, around 50% of the bee colonies had died. The hives of the dead colonies were stored at the beekeeper’s warehouse in the same conditions that he found them in the field, awaiting cleaning (Figure 1A).

Only a few dead bees were found in the brood chamber frames of these hives, (Figure 1B), with no anatomical deformities and no *Varroa* mites detected at the bottom of the hives or in worker bees and sealed brood. Moreover, there were no clinical signs of chalkbrood and American or European foulbrood. The beekeeper also conserved the acaricide strips used in the previous autumn in plastic bags (Figure 1C).

In the surviving colonies (Figure 1D), there were no more than two combs from the brood chamber that were covered with adult honey bees, much fewer than the five to seven combs that would be expected to be covered in this geographical area at that time (Figure 1E). The acaricide strips used in the autumn treatment were essentially untouched by the honey bees (Figure 1F), and there were no clinical signals of varroosis or other diseases in the brood or in adult worker honey bees. *Varroa* mites were not identified in brood cells or at the bottom of the hives.

The presence of accessible pollen and honey reserves in both the dead and surviving colonies ruled out death by starvation.

Most of the surviving colonies (approximately 200) did not have a large-enough adult population to ensure their future survival. 

Samples of worker honey bees and stored pollen were collected from the brood chamber from the dead and surviving colonies by veterinarians from the Centro de Investigación Apícola y Agroambiental’ (CIAPA) in Marchamalo (Guadalajara, Spain) to make a diagnosis of the causes of the collapsing event.

With the permission of the beekeeper, dead colonies (*n* = 5) and surviving colonies (*n* = 10) with sufficient honey bees in the frames were randomly selected to take around 300 worker honey bees per hive to study pathogens according to the methodology described in Section 4. In addition, four or five pieces of honey bee combs (10 × 15 cm each) that contained stored pollen were also randomly taken from different areas of the brood chamber from each colony surveyed in order to conduct a chemical and palynological analysis of the stored pollen (beebread). Finally, climatic parameters were also considered in the diagnostic analysis.

### 2.2. Pathogen Screening

In only one of the sampled dead colonies, there was a 5% parasitization by *V. destructor*, and the mite was not detected in any of the other hives. Of the surviving colonies sampled, only two were positive for *Varroa* mite with infestation rates of 25% and 1%. 

All the samples were positive for *N. ceranae*, with severe percentages of parasitization that were statistically higher in dead colonies (*p*-value = 0.001312). 

The presence of deformed wing virus (DWV) was confirmed in surviving colonies infected by *Varroa* and in six other samples. Moreover, black queen cell virus (BQCV) was present in six samples, one of which was positive for *V. destructor* (Table 1). Finally, *N. apis, Acarapis woodi*, Trypanosomatids, Neogregarines, Lake Sinai virus complex (LSV), and acute bee paralysis virus–Kashmir bee virus–Israeli acute paralysis virus complex (AKI) were not detected in any sample. 

#### *Varroa* Mite Resistance to Acaricides

Only 1 of the 15 colonies sampled (S1) complied with the criteria for conducting a resistance test [17]. The number of alive mites was lower (0, 1) than in control (7, 9) after 6 or 24 h of incubation, respectively. Therefore, it is concluded that this colony did not show any evidence of *Varroa* mite resistant to acaricides in any batch exposed to different acaricides (Table 2).

### 2.3. Stored Pollen Analysis

Of the 67 substances analyzed in beebread samples, only tau-fluvalinate and coumaphos were detected (Table 1), and there were statistically higher concentrations of tau-fluvalinate and coumaphos in the beebread samples from the surviving colonies (W = 43.5, *p*-value = 0.01312; W = 42.0, *p*-value = 0.02165, respectively; Table 1). The mean TUm value of the whole apiary was 0.00262 ± 0.00199. TUm values were <1 in both the dead and surviving colonies (Appendix A), indicating that the residue levels did not, in principle, reach the threshold of acute toxicity. Tau-fluvalinate represented less than 1% of TUm of the colonies (Appendix A). 

All the compounds identified in a given mixture contribute to TUm in accordance with their potency and the levels of their residues. Thus, as the levels of residues were significantly higher in the surviving colonies, TUm was also significantly higher in these hives (Appendix A). 

No significant differences were found between the dead and surviving colonies regarding the presence of wild flora in the beebread samples (W = 18.0, *p*-value = 0.21299). In four of the five samples from the dead colonies, wild plants were majorly present in the beebread. The most frequent taxa of wild plants identified were: *Araliaceae*, *Labiatae*, *Asteraceae*, *Chenopodiaceae*, and *Diplotaxis* spp. The major cultivated taxa detected was sunflower (*Helianthus annuus*, L.). In the surviving colonies, the predominant pollen was from wild plants in 6 out of 10 samples belonging to eight taxa: *Araliaceae*, *Labiatae*, *Caryophyllaceae*, *Cichorioideae*, *Convolvulaceae*, *Asteraceae*, *Chenopodiaceae*, and *Diplotaxis* spp. In the remaining 4 samples, the predominant taxa identified in the beebread were *H. annuus*, *Prunus* spp., and *Brassicaceae* (Appendix A).

### 2.4. Meteorological Data 

Figure 2a shows a Walter–Leith diagram (see Section 4.3 for details) of the meteorological station nearest the apiary for historical data (2013–2021). The mean annual temperature was 12.7 °C, and the yearly precipitation for this period of time was 485.3 mm. The seasonal distribution of the precipitation showed a maximum peak during the spring season and a secondary one during the fall–winter season. The dry season extends from June to September. The different bioclimatic indices estimated showed that the climate in the location corresponds to the eutemperate latitudinal belt, and it can be classified as oceanic-low continental. The bioclimate is classified as Mediterranean pluviseasonal-oceanic, within the low supramediterranean upper dry bioclimatic belt. 

In contrast, the year of the colony collapse event (2015) was characterized to be drier than the historical data (*p* = 266.4 mm), especially during the autumn–winter season, and it had a wider temperature amplitude between the coldest and hottest months (Figure 2b).

## 3. Discussion

Here we investigated a specific case study of the weakening and death of honey bee colonies in the field, a situation that has occurred quite frequently in Spain in recent years. The first suspected cause of colony death in this apiary was the action of the *Varroa* mite due to a failure in acaricidal treatment, often proposed to cause these effects [18]. However, the visual examination and pathogen screening suggested a different origin of the collapse, and in fact, *V. destructor* was detected in only 20% of the colonies sampled (3 out of 15). There were no clinical signs consistent with generalized varroosis in the apiary or in the hives in which the honey bee colonies died. Moreover, 2 of the positive colonies had a parasitic mite load of only 1% and 5%, not apparently representing an immediate risk to bee health [18]. Only 1 of the surviving honey bee colonies sampled (10%) had a higher parasite load (25%) and was thus at risk of suffering the negative effects of this mite [18]. After decades of miticide use against varroosis, there is a general concern about the selection of *Varroa* mites tolerant to acaricides [19]. However, the preliminary results of the acaricide resistance test and the absence of clinical signs of varroosis upon inspection suggest that resistant *Varroa* mites do not affect treatment efficacy in this apiary. Hence, it was concluded that the symptoms observed were not due to clinical varroosis after therapeutic failure. Overall, these results indicate that the *Varroa* mite might have exerted a degree of pressure on some individual colonies, but it is insufficient to provoke the collapse/weakening evident across the entire apiary.

By contrast, severe *N. ceranae* infection was detected in all cases, especially in dead colonies, with clinical signs and symptoms in all cases (dead and surviving) in line with the infections observed previously in colonies that collapsed in winter due to such infestation [20]. The dynamics of nosemosis C in the colony provokes the mean spore count to fluctuate greatly from the start to the end of the disease in interior bees, and it is not a reliable measure of a colony’s health when bees are infected with *N. ceranae* [20]. In fact, the proportion of foraging bees infected with *N. ceranae* was the strongest indicator of the spread of the disease in the colony. In this sense, forager bees are always more infected than house bees. The more foragers that are infected, the smaller size of the bee colony. However, percentages of parasitization by *N. ceranae* in house bees higher than 35%–40% indicate a serious risk for the colony of suffering from nosemosis C, which could reach collapse [21]. Moreover, as expected, a lower infestation was seen in early spring, as described in the phases 3 and 4 of the disease [20] and consistent with its evolution in different seasons [20,22,23]. The fact that the acaricide strips did not change color or underwent propolization probably reflects a change in honey bee behavior due to *N. ceranae* infection, which could produce a serious risk that the acaricide treatment would lose efficacy if the *V. destructor* mites were abundant [23,24]. Indeed, *N. ceranae* alters various physiological processes in individual honey bees involving immunomodulation [25,26,27] and energetic stress [28,29], inducing early foraging activities [30,31]. These alterations have a direct impact on the colony [20,21,32], especially in geographical areas with warmer climates where there is a large concentration of professional beekeeping [33,34,35], in contrast to colder climates ([36,37,38,39] reviewed in [23]).

Although there was a scattered presence of BQCV and DWV in the samples, their detection may be a consequence of a side effect of the presence of *Varroa* mites [18,40,41,42] and *Nosema* spp. [40,41,43,44]. The viral loads were not determined because workers did not display DWV clinical signs. The latent presence of viruses in *Apis mellifera* is well known in the literature, and while showing no signs of disease, they may destroy bee fitness and health during favorable conditions (e.g., *V. destructor* infestations). However, in the case of infection by *N. ceranae*, it has been demonstrated that these two pathogens are not acting synergistically [45,46,47,48]. Moreover, under laboratory conditions, it has been observed that the inoculation of DWV does not have an impact on *N. ceranae* infection. On the contrary, prior establishment of *N. ceranae* has a significant negative impact on the load of DWV [49].

Neonicotinoids and other agrochemicals were not detected in the beebread, consistent with the fact that honey bees mainly visited wild flora. Nevertheless, the high concentrations of tau-fluvalinate and coumaphos detected in beebread samples were assumed to have a beekeeping origin as these chemicals are registered in Spain to control *Varroa* mite. Moreover, tau-fluvalinate and coumaphos have octanol, water partitioning coefficient (logKow) > 3 [50,51], indicating high lipophilicity and potential to accumulate in wax [52,53] and other hive matrices [54], where they may remain relatively stable for long periods of time [55,56,57]. Indeed, both of these acaricides are estimated to need 5 years to completely disappear from bee matrices [58]. Moreover, their concentration in wax may increase due to the wax recycling processes [55,56], explaining why their residues are frequently found in wax and beebread worldwide [53,59,60,61]. If the acaricide residue levels in beebread reaches toxic levels, the health of the honey bee colony might be compromised. Thus, a synergetic toxic effect between these acaricides cannot be ruled out. Indeed, acute contact toxicity of coumaphos increased up to 3- to 4-fold when 4-day-old honey bees were pretreated with tau-fluvalinate at a dose of 1 or 3 µg/bee, and the contact toxicity of tau-fluvalinate increased up to 32-fold when the individuals were pretreated with coumaphos at a dose of 10 µg/bee [62]. 

In addition, high concentrations of acaricides may have made the honey bee colonies more sensitive to *N. ceranae* infection [63]. In this sense, following the toxic unit (TU) approach and based on acute toxicity, LC50, it has been proposed that Ln (TU) = −6.706 may represent as a preliminary break point regarding the increment of *N. ceranae* when assessed in the presence of a mixture of xenobiotics [14]. The mean Ln(TUm) in the present apiary was −5.95, suggesting that it may have been more vulnerable to *N. ceranae* infection [63]. However, Ln(TUm) values were higher in surviving colonies, which also had lower *N. ceranae* infection. This may lead to the erroneous conclusion that a high miticide concentration contributes to colony survival. However, this was not the case because their viability was compromised in early spring due to the small adult honey bee population. Thus, while more than 50% of the dead honey bees are expected to be infected in colonies that collapsed due to nosemosis C in the cold months, this percentage is lower when colonies collapse later in the year, probably due to an increment in the proportion of uninfected newborn honey bees [20].

Finally, the unusual climatic conditions of the year of the collapse event may have had an influence on the strength of the colonies. On one side, the warmer and drier conditions of this year may have provoked a change in the phenology and physiology of the vegetation of the zone, with a decrease in the length of the flowering period and, therefore, and quality of the collected pollen [64], which may indirectly have affected the strength of the colonies [65,66]. In this direction, the impact of weather conditions on the overwintering survival of colonies has recently been studied in Pennsylvania for 3 years [67]. Despite the short database, the authors found adverse effects of both too-cool and too-hot summer on overwintering colony survival [67]. These results are in line with the ones found in other countries [67]. In addition, these climatic conditions may have favored the infection with *N. ceranae*, whose spores resist high temperatures and desiccation, and they complete their life cycle more efficiently at high temperatures [68].

## 4. Material and Methods

### 4.1. Pathogen Screening

The number of bees present in each sample (around 300 bees) was counted, and the bees were examined individually to detect the presence of *Varroa* mites and collect them by means of sterile tweezers [14]. Each mite detected was analyzed macroscopically to confirm the species. A honey bee colony was considered infested with *V. destructor* when at least 1 *Varroa* mite was found in the sample. The rate of infestation of the bee colony was estimated by assessing the number of *Varroa* mites in relation to the number of adult bees in each sample, and it was expressed as the number of *Varroa* mites/100 bees/sample [69].

The presence of *Nosema* spp., Trypanosomatids, Neogregarines, and *Acarapis woodi* was evaluated in a sample (*n* = 60) from each colony. The remaining bees were kept frozen at −80 °C. 

The presence of different viruses was only analyzed in the surviving colonies sampled because the viral RNA integrity could not be ensured in the dead colonies.

The subsample of each colony (*n* = 60) was macerated in 50% AL buffer (Qiagen GmbH, Hilden, Germany) before DNA and RNA were extracted, as detailed in [69,70,71]. 

Briefly, macerated bees were centrifuged at 3000 rpm for 10 min, and the resulting pellets were used for DNA extraction, and the supernatants for RNA extraction. Both the pellets and supernatants were stored at −80 °C prior to nucleic acid (DNA or RNA) extraction. For DNA extraction, the pellets were resuspended in 3 mL MilliQH_2_O, and a 400 μL aliquot was transferred to a 96-well plate (Qiagen) with glass beads (2 mm diameter, Merck KGaA, Darmstadt, Germany) using disposable Pasteur pipettes. After overnight preincubation with proteinase K (20 μL, Qiagen), the samples were then processed as described previously [71] following the BS96 DNA Tissue extraction protocol in a BioSprint station (Qiagen). The plates were then stored at −20 °C. For RNA extraction, 400 μL of the supernatant was incubated for 15 min with protease (20 μL, Qiagen) at 70 °C, and the nucleic acids were then extracted as described above (BioSprint 96 DNA in, BioSprint workstation, Qiagen). The total nucleic acids recovered were then subjected to DNA digestion with DNase I (Qiagen) to completely remove any genomic DNA, and the total RNA recovered was used immediately to generate firststrand cDNAs using the QuantiTect Reverse Transcription Kit (Qiagen) according to the manufacturer’s instructions. The resultant cDNA was used for subsequent virus analysis with no further dilution [70]. 

Negative and positive controls were run in parallel for each step: bee maceration, DNA and RNA extraction, and reverse transcription [69].

Published PCR or RT-PCR protocols were considered to screen *Nosema apis* and *N. ceranae* [71], Trypanosomatids and Neogregarines [72,73], *Acarapis woodi* [74], LSV complex [75], AKI complex [76], DWV, and BQCV [77]. Appendix A includes the primers used.

In addition, the proportion of *Nosema* spp. infection was determined by PCR on 25 individual worker honey bees from each colony sampled [71]. 

#### Test of *Varroa* Mite Resistance to Acaricides

When possible, mite resistance to acaricides was determined using the respective marketed products, CheckMite^®^ (a.m.: coumaphos), Apistan^®^ (a.m.: tau-fluvalinate), and Apitraz^®^ (a.m.: amitraz), according to the protocol described previously [17] with the following modifications:
Inclusion of an additional batch for a 24 h incubation period;Feeding the honey bees with syrup during the incubation periods; andFreezing the honey bees at the end of the incubation period to collect the remaining *Varroa* mites (−80 °C, 15 min).

The honey bees were kept at 35 °C during the test, and after incubation periods, a control test without treatment was used to determine how the basal conditions affected *Varroa* mite mortality.

### 4.2. Stored Pollen Analysis

Beebread was extracted aseptically from the combs, removing the wax and preparing a composite sample for each colony. Finally, each pollen sample was divided into two 100 g aliquots, for chemical and palynological analyses, and stored at −80 °C.

A multiresidue chemical analysis of 60 substances was carried out following a method described elsewhere [59], assessing acaricides (AC), fungicides (FU), herbicides (HB), and insecticides (IN). In addition, 7 neonicotinoid INs (acetamiprid, clothianidin, dinotefuran, imidacloprid, nitenpyram, thiacloprid, and thiamethoxam) were measured as described previously [78].

Based on the results of the multiresidue analysis and the toxicity data reported previously (Table A2 in [14]), the toxic unit of the mixtures (TUm) was calculated following an approach given elsewhere [14] to assess the risk of the chemical mixture found in each hive sampled. Subsequently, the natural logarithm (Ln(TUm)) was estimated for comparison purposes with [14]. In that work, the possible relationships between TUm and the prevalence of pathogens were studied by using a factor analysis. The natural logarithm of TUm, Ln(TUm), was derived to normalize data. Thus, the higher the value of Ln(TUm), the higher the toxicity risk.

Finally, the type of foraging flora was confirmed by analyzing beebread samples as described elsewhere [11,20,21] and estimating the proportion of pollen from wild (WP) and cultivated (CP) plants. The pollen of the beebread was extracted by diluting 0.5 g in 10 mL of acidulated water (0.5% sulfuric acid) and centrifuging at 2500 r.p.m. for 15 min. The pellet was washed with double-distilled water and centrifuged twice. The sediment was placed onto a glycerine jelly slide and examined microscopically in order to identify the pollen. The frequency of the pollen grains of each taxon is expressed as a percentage of the total pollen grains. Between 300 and 1200 pollen grains were counted in each sample.

The pollen grains were identified and classified on the basis of the identification keys [79,80] and the pollen slide reference collection available at the honey laboratory at the CIAPA.

### 4.3. Meteorological Data

A Walter–Leith diagram [81] was developed with historical weather data obtained from the meteorological station of Brihuega (lat. 40.765, long. −2.874) from the network of the State Meteorological Agency (AEMET) to compare it with the data for the period of the study. This weather station is located 9 km far away from the studied areas. 

Walter and Leith climate diagrams are brief summaries of average climatic variables and their time course. They illustrate precipitation and temperature changes throughout the year in 1 standardized chart. Originally aimed at visualizing those climatic variables and their dynamics, which are particularly important for vegetation, they have proven useful for a wide range of sciences. The diagrams were developed with the diagnostic tool of the Worldwide Bioclimatic Classification System, 1996–2021 [82].

### 4.4. Statistical Analysis

A 1-tailed Mann–Whitney test (α = 0.05) was used to analyze possible differences between the dead and surviving colonies in terms of the different experimental parameters measured (pathogens, % wild pollen in beebread, chemical residues, and TUm). The analysis was carried out with Statgraphics Centurion 18^©^.

## 5. Conclusions

The veterinary inspection and analytical evidence presented here indicate that nosemosis C infection was the underlying cause of the colony weakness and collapse of the professional apiary studied, probably accelerated by the presence of high levels of miticides and unusual climatic conditions. In conjunction with the unchecked concentrations of acaricide that accumulated in honey bee hives, *N. ceranae* infection represents a real danger in honey bee colony survival. Therefore, in addition to the correct use of veterinary products to control *V. destructor*, appropriate wax renewal of the combs should be introduced to develop specific preventive strategies aimed at controlling possible infections from prevalent pathogens.

## Figures and Tables

**Figure 1 pathogens-10-00955-f001:**
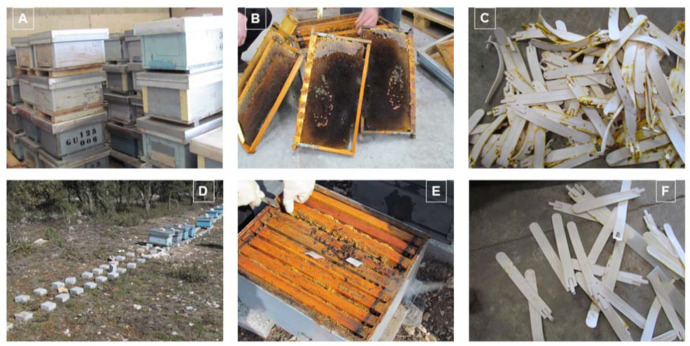
Pictures obtained during the inspection of colonies. (**A**) Hives stored in the beekeeper’s warehouse; (**B**) brood combs from the dead colonies with a few honey bees and sealed brood in the frames; (**C**) CheckMite^®^ strips kept by the beekeeper after their removal; (**D**) apiary with many empty spots and a few surviving colonies; (**E**) weak surviving colony with a small honey bee population; (**F**) details of some CheckMite^®^ strips with low interaction with honey bees.

**Figure 2 pathogens-10-00955-f002:**
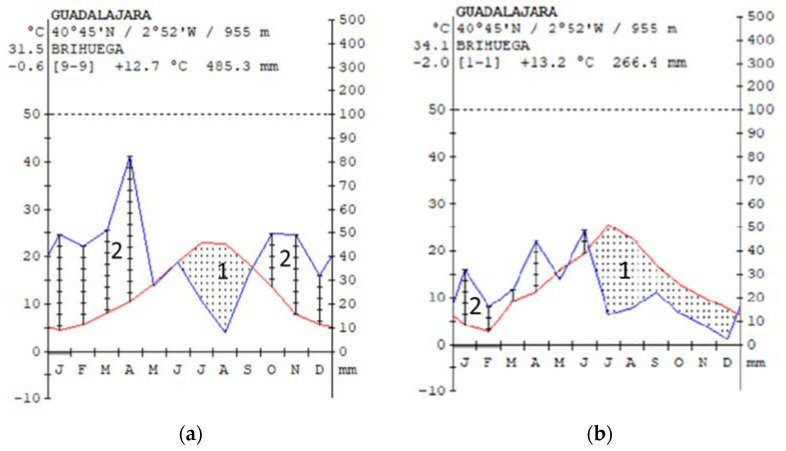
Climatic diagram drawn according to Walter and Leith (1960) for (**a**) the period 2013–2021 and (**b**) the studied year (2015) data of the Brihuega station. (**------**) monthly mean temperature (°C); (**-----**) monthly precipitation. The plot shows 1 = dry season and 2 = rainy season.

**Table 1 pathogens-10-00955-t001:** Results of pathogen and pesticide residue screening and palynological analysis of beebread samples from dead (D) and weak yet surviving (S) colonies. Pathogen screening: *V. destructor* parasitization (% VD), prevalence of *N. ceranae* (% NC), and detection (+ or −) of deformed wing virus (DWV) and black queen cell virus (BQCV). Residues quantified in beebread (ppb): tau-fluvalinate (FVT) and coumaphos (CMF). Natural logarithm of the toxic unit of the mixture in each colony (Ln(TUm)). Palynological analysis: percentage of wild foraging plants (%WP). Grey cells in the table indicate parameters not analyzed.

Status of the Colonies	Colony Code	%VD	%NC *	DWV	BQCV	FVT ^(*)^	CMF *	LN(TUm) *	%WP
Dead	D1	5	70			7	435	−6.38	96.3
	D2	0	80			7	415	−6.43	45.1
	D3	0	75			<LOQ	202	−7.154	89.3
	D4	0	92			13	350	−6.59	65.1
	D5	0	89			9	323	−6.68	79.4
Surviving	S1	25	20	+	+	7	283	−6.81	90.5
	S2	1	30	+	−	9	545	−6.16	30.7
	S3	0	25	+	−	10	2230	−4.75	15.2
	S4	0	60	+	+	15	465	−6.31	30.5
	S5	0	36	+	+	20	1165	−5.42	33.7
	S6	0	20	+	−	16	305	−6.73	77.7
	S7	0	30	−	+	18	775	−5.80	92.1
	S8	0	45	−	+	19	850	−5.71	55.6
	S9	0	35	+	−	13	936	−5.62	85.4
	S10	0	35	+	+	13	845	−5.72	93.7

* Statistically significant differences between dead and surviving colonies at α = 0.05. LOQ: level of quantification.

**Table 2 pathogens-10-00955-t002:** Results of the screening to identify *Varroa mites* from the S1 colony resistant to acaricides.

Incubation Time	Mites	Control	CheckMite^®^ (Coumaphos)	Apistan^®^ (Tau-Fluvalinate)	Apitraz^®^ (Amitraz)
6 h	Dead	1	10	5	8
	Alive	7	0	0	0
24 h	Dead	2	11	7	8
	Alive	9	0	1	1

## Data Availability

The data presented in this study are available on request from the corresponding author.

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
