# Peer review of "A Case Report of Chronic Stress in Honey Bee Colonies Induced by Pathogens and Acaricide Residues"

_pathogens, 2021, doi:10.3390/pathogens10080955_

Round 1
Reviewer 1 Report
The manuscript entitled "A case report of chronic stress in honey bee colonies induced by pathogens and acaricide residues". This study the authors analyze the possible causes of the poor health status of a professional apiary in Spain. They evaluated potential factors associated to this poor health status, these factors including Nosema ceranae infection, common viruses (eg, BQCV and DWV infection), and the accumulation of acaricides commonly used to control Varroa destructor in the beebread (coumaphos and tau-fluvalinate). Based on these analyses, they suggest that the main factor in the colony weakening and dead was due to N. ceranae, helped by the presence of high levels of acaricides. I cannot deny that N. ceranae may be an important factor in colony loss, but I believe that other pathogens, such as DWV and trypanosomes are also important factors in colony loss; factors that are not well addressed or analyzed in this study. Therefore, the authors must include new information to improve this study. For specific comments to improve this manuscript, please see below.
- 50. etc change by others
- 67-70. I recommend moving this paragraph in materials and methods section.
- 104. infection by infestation
- 105. How did you calculate the % parasitization?. I believe there is more informative to include the number of spores per bees.This information allows us to observe the degree or level of infection that the infected bees can show.
- 107. …The presence of Deformed Wing Virus (DWV) was confirmed in surviving colonies infected by Varroa and in six other samples… Here I have my apprehensions with this result. The positive or negative case only indicates the presence/absence of DWV in the sample, but this information no indicates the viral infection level. DWV was detected in high or low load? How many copies per bees were detected?
Currently, there also reported at least three variants of DWV (DWV-A, DWV-B and DWV-C) are found in honey bees. Which one was detected in this study?. Now we know that DWV is important factor in the honey bee decline. Therefore, I have my doubts that N. ceranae was the only factor in the death and weakening of the colonies reported in this study. More information about the Nosema ceranae and DWV loads (and the other viruses detected) needs to be included in the results section.
- 110. Acarapis. woodi. Delete dot between Acarapis woodi
- 110. I am not sure that trypanomatids/Neogregarines were absent in the samples since the primers designed/used by "Meeus et al. 2010. Multiplex PCR detection of slowly-evolving trypanosomatids and neogregarines in bumblebees using broad-range primers" are applicable for specific detection of trypanomatids/Neogregarines in honey bees. These primers are recommended for specific detection of trypanomatids/Neogregarines that parasitize Bombus spp, such as Crithidia bombi and Apicystis bombi. Of course, there exist that possibility that these pathogen could be found also in honey bees, but they are not so common in these bees. Therefore, I believe the results reported (trypanomatids/Neogregarines detection) in this study could be false negative.
There are reported other specific primers for detection e.g. trypanosomatids, such as Lotmaria passim and Crithidia mellificae, which are more frequent in honey bees. I recommend analyzing the bee samples with primers report by e.g. Arismendi et al. 2016 (https://doi.org/10.1016/j.jip.2015.12.008), Stevanovic et al. 2016 https://doi.org/10.1016/j.jip.2016.07.001) or Xu et al., 2018 (https://doi.org/10.1007/s00436-017-5733-2). I believe the trypanosomes detection with new primers must be must be update in the Results section.
- 113-117. Title Table 1. Please write the scientific name in italic. Also, in the Table 1 was absent the results associated to detection of N. apis (% NA), (+ or -) of A. woodi (AW), Trypanosomatids (Tryp), Neogregarines (Neog), Lake Sinai Virus complex complex (LSV), and Acute Bee Paralysis Virus-Kashmir Bee Virus Israeli Acute Paralysis complex. The authors stated in title of table 1 that these results are reported in the table. See title Table 1. “Results of pathogen and pesticide residue screening and the palynological analysis of beebread samples from dead (D) and weak yet surviving (S) colonies. Pathogen screening: V. destructor parasitization (% VD), prevalence of N. ceranae (% NC) and N. apis (% NA) and detection(+ or -) of A. woodi (AW), Trypanosomatids (Tryp), Neogregarines (Neog), Lake Sinai Virus complex complex(LSV), Acute Bee Paralysis Virus-Kashmir Bee Virus Israeli Acute Paralysis, Virus complex (AKI), Deformed Wings Virus (DWV) and Black Queen Cell Virus (BQCV)”...
Please to correct this information.
- 174 infestation by infection
L 241. I recommend using new primers for trypanosomes detection or/and quantification (e.g. Arismendi et al. 2016 (https://doi.org/10.1016/j.jip.2015.12.008), Stevanovic et al. 2016 https://doi.org/10.1016/j.jip.2016.07.001), Xu et al., 2018 (https://doi.org/10.1007/s00436-017-5733-2). Primers reported by Meeus et al. (2010) are not adequate for detection of common trypanosomes in honey bees.
- 282-284. “The veterinary inspection and analytical evidence presented here indicate that nosemosis C infection was the underlying cause of the colony weakness and collapse at the professional apiary studied”.
I am not sure if N. ceranae was the main causal agent of colony weakness and colonies collapse in study. I would like to see the information in N. ceranae, DWV and trypanosomes load level in honey bee collected. This information must be showed in this study.
Author Response
Answer to Reviewer 1: In the literature is well described the spore counts are not a good predictor of the evolution of the nosemosis C in the colonies, but the percentage of foragers and home bees infected [1] . The dynamic of nosemosis C in the colony provokes the mean spore count fluctuates greatly from the start to the end of the disease in interior bees and it is not a reliable measure of a colony’s health when bees are infected with N. ceranae. Thus, foragers are always more infected than interior bees. The more foragers infected, the smaller the number of brood combs and the fewer frames of bees. On the other hand, DWV loads were not determined because clinic signs of virosis were not detected during the veterinary visit. In previous works, it is demonstrated these two pathogens are not acting synergistically [2-5]. Furthermore, under laboratory conditions, it has been observed the inoculation of DWV does not have an impact on N. ceranae infection. On the contrary, prior establishment of N. ceranae has a significant negative impact on the load of DWV [6].
Regarding the doubts on the primers used for the detection trypanosomatids and neogregarines, we are aware there are more specific primers in the literature. Indeed our group has recently developed specific primers and massive sequencing technology for detection nosematid¸ trypanosomatid and neogregarine species [7,8] with the aim of improving the knowledge of the pathogeny of emergent pathogens, especially trypanosomatids [8,9] . However, we decided to use Meuus’ primers with screening purposes, as made in previous works [10-12], due to their high sensitiveness. In any case, following the recommendations of the reviewer, we used Stevanovic et al primers [13] in order to confirm the negative results and the manuscript updated accordingly. Finally, the typos in the text were all corrected
References
- Higes, M.; Martín-Hernández, R.; Botías, C.; Bailón, E.G.; González-Porto, A.V.; Barrios, L.; del Nozal, M.J.; Bernal, J.L.; Jiménez, J.J.; Palencia, P.G.; et al. How natural infection by Nosema ceranae causes honeybee colony collapse. Environmental Microbiology 2008, 10, 2659-2669, doi:https://doi.org/10.1111/j.1462-2920.2008.01687.x.
- Costa, C.; Tanner, G.; Lodesani, M.; Maistrello, L.; Neumann, P. Negative correlation between Nosema ceranae spore loads and deformed wing virus infection levels in adult honey bee workers. Journal of Invertebrate Pathology 2011, 108, 224-225, doi:https://doi.org/10.1016/j.jip.2011.08.012.
- Dussaubat, C.; Brunet, J.-L.; Higes, M.; Colbourne, J.K.; Lopez, J.; Choi, J.-H.; Martín-Hernández, R.; Botías, C.; Cousin, M.; McDonnell, C.; et al. Gut Pathology and Responses to the Microsporidium Nosema ceranae in the Honey Bee Apis mellifera. PLOS ONE 2012, 7, e37017, doi:10.1371/journal.pone.0037017.
- Hedtke, K.; Jensen, P.M.; Jensen, A.B.; Genersch, E. Evidence for emerging parasites and pathogens influencing outbreaks of stress-related diseases like chalkbrood. Journal of Invertebrate Pathology 2011, 108, 167-173, doi:https://doi.org/10.1016/j.jip.2011.08.006.
- Martin, S.J.; Hardy, J.; Villalobos, E.; Martín-Hernández, R.; Nikaido, S.; Higes, M. Do the honeybee pathogens Nosema ceranae and deformed wing virus act synergistically? Environmental Microbiology Reports 2013, 5, 506-510, doi:https://doi.org/10.1111/1758-2229.12052.
- Doublet, V.; Natsopoulou, M.E.; Zschiesche, L.; Paxton, R.J. Within-host competition among the honey bees pathogens Nosema ceranae and Deformed wing virus is asymmetric and to the disadvantage of the virus. Journal of Invertebrate Pathology 2015, 124, 31-34, doi:https://doi.org/10.1016/j.jip.2014.10.007.
- Bartolomé, C.; Buendía-Abad, M.; Benito, M.; Sobrino, B.; Amigo, J.; Carracedo, A.; Martín-Hernández, R.; Higes, M.; Maside, X. Longitudinal analysis on parasite diversity in honeybee colonies: new taxa, high frequency of mixed infections and seasonal patterns of variation. Scientific Reports 2020, 10, 10454, doi:10.1038/s41598-020-67183-3.
- Buendía-Abad, M.; Higes, M.; Martín-Hernández, R.; Barrios, L.; Meana, A.; Fernández Fernández, A.; Osuna, A.; De Pablos, L.M. Workflow of Lotmaria passim isolation: Experimental infection with a low-passage strain causes higher honeybee mortality rates than the PRA-403 reference strain. International Journal for Parasitology: Parasites and Wildlife 2021, 14, 68-74, doi:https://doi.org/10.1016/j.ijppaw.2020.12.003.
- Buendía-Abad, M.; García-Palencia, P.; Pablos-Torró, L.M.d.; Alunda, J.M.; Osuna, A.; Martín-Hernández, R.; Higes, M. First description of Lotmaria passim and Crithidia mellificae haptomonad stage in the honeybee hindgut. bioRxiv 2021.
- Alonso-Prados, E.; Muñoz, I.; De la Rúa, P.; Serrano, J.; Fernández-Alba, A.R.; García-Valcárcel, A.I.; Hernando, M.D.; Alonso, Á.; Alonso-Prados, J.L.; Bartolomé, C.; et al. The toxic unit approach as a risk indicator in honey bees surveillance programmes: A case of study in Apis mellifera iberiensis. Sci Total Environ 2020, 698, 134208, doi:10.1016/j.scitotenv.2019.134208.
- Gómez-Moracho, T.; Buendía-Abad, M.; Benito, M.; García-Palencia, P.; Barrios, L.; Bartolomé, C.; Maside, X.; Meana, A.; Jiménez-Antón, M.D.; Olías-Molero, A.I.; et al. Experimental evidence of harmful effects of Crithidia mellificae and Lotmaria passim on honey bees. International Journal for Parasitology 2020, 50, 1117-1124, doi:https://doi.org/10.1016/j.ijpara.2020.06.009.
- Cepero, A.; Ravoet, J.; Gómez-Moracho, T.; Bernal, J.L.; Del Nozal, M.J.; Bartolomé, C.; Maside, X.; Meana, A.; González-Porto, A.V.; de Graaf, D.C.; et al. Holistic screening of collapsing honey bee colonies in Spain: a case study. BMC Research Notes 2014, 7, 649, doi:10.1186/1756-0500-7-649.
- Stevanovic, J.; Schwarz, R.S.; Vejnovic, B.; Evans, J.D.; Irwin, R.E.; Glavinic, U.; Stanimirovic, Z. Species-specific diagnostics of Apis mellifera trypanosomatids: A nine-year survey (2007–2015) for trypanosomatids and microsporidians in Serbian honey bees. Journal of Invertebrate Pathology 2016, 139, 6-11, doi:https://doi.org/10.1016/j.jip.2016.07.001.
Annex: detailed answer to each comment.
- etc change by others.
The intro has been updated expanding the causes of the CDD phenomenom.
67-70. I recommend moving this paragraph in materials and methods section.
We consider this recommendation may make the text does not have an unifying threat. We prefer to keep this paragraph in order to keep the context with the next one . Nevertheless, we moved the heading of sampling at the end of the veterinary visit heading.
- infection by infestation.
Done
- How did you calculate the % parasitization?. I believe there is more informative to include the number of spores per bees. This information allows us to observe the degree or level of infection that the infected bees can show.
It is described in the point 4.2 of the section of Material and Methods (lines 243-244): “In addition, the proportion of Nosema spp. infection was determined by PCR on 25 individual worker honey bees from each colony subsample”
The dynamic of nosemosis C in the colony provokes the mean spore count fluctuates greatly from the start to the end of the disease in interior bees and it is not a reliable measure of a colony’s health when bees are infected with N. ceranae. Thus, foragers are always more infected than interior bees. The more foragers infected, the smaller the number of brood combs and the fewer frames of bees. In fact, the proportion of forager bees infected with N. ceranae was the only useful indicator of the extent of disease in the colony [1].
- …The presence of Deformed Wing Virus (DWV) was confirmed in surviving colonies infected by Varroa and in six other samples… Here I have my apprehensions with this result. The positive or negative case only indicates the presence/absence of DWV in the sample, but this information no indicates the viral infection level. DWV was detected in high or low load? How many copies per bees were detected?
Currently, there also reported at least three variants of DWV(DWV-A, DWV-B and DWV-C) are found in honey bees. Which one was detected in this study?. Now we know that DWV is important factor in the honey bee decline. Therefore, I have my doubts that N. ceranae was the only factor in the death and weakening of the colonies reported in this study. More information about the Nosema ceranae and DWV loads (and the other viruses detected) needs to be included in the results section.
The viral loads were not determined because workers did not display DWV clinical signs. The latent presence of virus in Apis mellifera is well known in the literature, and, while showing no signs of disease, they may destroy bee fitness and health during favourable conditions (e.g. Varroa destructor infestations). However, in the case of the infection by Nosema ceranae, it has been demonstrated these two pathogens are not acting synergistically [2-5]. Moreover, under laboratory conditions, it has been observed the inoculation of DWV does not have an impact on N. ceranae infection. On the contrary, prior establishment of N. ceranae has a significant negative impact on the load of DWV [6].
- Acarapis. woodi. Delete dot between Acarapis woodi
Done
- I am not sure that trypanomatids/Neogregarines were absent in the samples since the primers designed/used by "Meeus et al. 2010. Multiplex PCR detection of slowly-evolving trypanosomatids and neogregarines in bumblebee susing broad-range primers" are applicable for specific detection of trypanomatids/Neogregarines in honey bees. These primers are recommended for specific detection of trypanomatids/Neogregarines that parasitize Bombus spp,such as Crithidia bombi and Apicystis bombi. Of course, there exist that possibility that these pathogen could be found also in honey bees, but they are not so common in these bees. Therefore, I believe the results reported (trypanomatids/Neogregarines detection) in this study could be false negative.
There are reported other specific primers for detection e.g. trypanosomatids, such as Lotmaria passim and Crithidia mellificae, which are more frequent in honey bees. I recommend analyzing the bee samples with primers report by e.g. Arismendiet al. 2016 (https://doi.org/10.1016/j.jip.2015.12.008), Stevanovic et al. 2016 https://doi.org/10.1016/j.jip.2016.07.001) or Xu et al.,2018 (https://doi.org/10.1007/s00436-017-5733-2). I believe the trypanosomes detection with new primers must update in the Results section.
Meuus’ primers are very sensitive but no specific and we used them with screening purposes as in our previous works. In any case we repeated the PCR with Stevanovic’s primers confirming the absence of these pathogens.
The Material and Method and Results sections have been updated, accordingly.
113-117. Title Table 1. Please write the scientific name initalic. Also, in the Table 1 was absent the results associated to detection of N. apis (% NA), (+ or -) of A. woodi (AW), Trypanosomatids (Tryp), Neogregarines (Neog), Lake SinaiVirus complex complex (LSV), and Acute Bee Paralysis Virus-Kashmir Bee Virus Israeli Acute Paralysis complex. The authors stated in title of table 1 that these results are reported in the table. See title Table 1. “Results of pathogen and pesticide residue screening and the palynologicalanalysis of beebread samples from dead (D) and weaktsurviviny (S) colonies. Pathogen screening: V. destructorparasitization (% VD), prevalence of N. ceranae (% NC) andN. apis (% NA) and detection(+ or -) of A. woodi (AW),Trypanosomatids (Tryp), Neogregarines (Neog), Lake SinaiVirus complex complex(LSV), Acute Bee Paralysis Virus-Kashmir Bee Virus Israeli Acute Paralysis, Virus complex(AKI), Deformed Wings Virus (DWV) and Black Queen CellVirus (BQCV)”... Please to correct this information.
The table was reduced in order to fit the outline of the editorial. The no detection of these pathogens is already mentioned in the text. The inclusion of this information in the table is redundant.
The head of the table 1 has been updated
174 infestation by infection
Done
L 241. I recommend using new primers for trypanosomes detection or/and quantification (e.g. Arismendi et al. 2016(https://doi.org/10.1016/j.jip.2015.12.008), Stevanovic et al. 2016 https://doi.org/10.1016/j.jip.2016.07.001), Xu et al., 2018 (https://doi.org/10.1007/s00436-017-5733-2). Primers reported by Meeus et al. (2010) are not adequate for detection of common trypanosomes in honey bees.
Meuus’ primers were used with screening purposes. Following, the reviewer’s recommendations we confirmed our results with Stevanovic et al primers [13].
282-284. “The veterinary inspection and analytical evidence presented here indicate that nosemosis C infection was the underlying cause of the colony weakness and collapse at the professional apiary studied”.
I am not sure if N. ceranae was the main causal agent of colony weakness and colonies collapse in study. I would like to see the information in N. ceranae , DWV and trypanosomes load level in honey bee collected. This information must be showed in this study.
As mentioned before:
1.- The dynamic of nosemosis C in the colony provokes the mean spore count fluctuates greatly from the start to the end of the disease in interior bees and it is not a reliable measure of a colony’s health when bees are infected with N. ceranae. Thus, foragers are always more infected than interior bees. The more foragers infected, the smaller the number of brood combs and the fewer frames of bees [1].
2.- Regarding DWV, the viral loads were not determined because workers did not display DWV clinical signs. The latent presence of viruses in Apis mellifera is well known in the literature, and, while showing no signs of disease, they may destroy bee fitness and health during favourable conditions (e.g. Varroa destructor infestations). However, in the case of the infection by Nosema ceranae, it has been demonstrated these two pathogens are not acting synergistically [2-5]. Moreover, it has been observed under laboratory conditions, that the inoculation of DWV does not have an impact on N. ceranae infection. On the contrary, prior establishment of N. ceranae has a significant negative impact on the load of DWV [6].
3.- we used Meuus’ primers with screening purposes, due to their high sensitiveness as made in previous works [10-12]. In any case, following the recommendations of the reviewer, in order to confirm the negative results and we used Stevanovic et al primers [13] and the manuscript updated with the results.
Reviewer 2 Report
Elena Alonso-Prados and colleagues investigated potential causes of colony collapse in a Spanish apiary.
Line 38: about "maintaining the homeostasis of different ecosystems", I believe there are more adequate literatures to cite, such as:
Ernest SKM and Brown JH. 2001. Homeostasis and Compensation: The Role of Species and Resources in Ecosystem Stability. Ecology, 82(8):2118-2132.
In lines 48-54: the paragraph about the decline of honey bee colonies (and the different factors currently known and associated with CCD) should be better organized and expanded.
In item 2.1. Veterinary inspection: please write more clearly about the seasons and their corresponding months (beginning of autumn 2015, winter, early spring 2016).
In item 2.2. Pathogen screening: it is really important to mention what was not detected in the samples (lines 109-112), but I recommend not putting the undetected parameters in the legend of Table 1.
In lines 118-119: "natural logarithm of the toxic unit of the mixture in each colony (Ln(TUm))" needs to be better explained somewhere in the manuscript.
In item 2.1.1. Varroa mite resistance to acaricides (lines 122-126 and Table 2): the entire dataset needs to be clearly and better explained.
The discussion section is interesting, but it could be more objective (summarized) and focused only on the case study data.
The methodology is quite extensive but I still think it deserves a little more detail so that readers from other areas can understand the entire study.
I recommend that authors add more collected data as supplementary material, such as PCR or RT-PCR images, pollen images, and so on.
Author Response
Answer to Reviewer 2: the different causes involved in the CDD phenomenon has been expanded and reorganized in the introduction section. Additionally, further details on the test of Varroa mite resistance and Ln(TU) have been included in the results section. Regarding the discussion section, we focused on the possible cause of the collapse event based on the experimental results of screening analysis of pathogens, residue and palynological analysis of the beebread and climatic conditions, and justified why. In principle the beekeeper considered the main cause was the action of Varroa mites, which was ruled out due to the lack of symptomatic signs during the veterinary visit. Nevertheless, the infection levels by N. ceranae were very high which seem to be the main cause of the collapse of the apiary and the clinical signs. The palynological analysis showed the colonies foraged mainly on wild flora, which was not contaminated by pesticides used in the agriculture, ruling out the exposure to pesticides used in agriculture as possible driver of the collapse event. We supposed that the evolution of N. ceranae infection could be favoured by the presence of high levels of veterinary acaricides in the beebread and the unusual climatic conditions. Moreover, the discussion includes some information on how these factors can favour the evolution of the disease. Finally, further details on the methodology followed for the screening of pathogens are included in the ms and supplementary information.
Annex: detailed answer to each comment.
Line 38: about "maintaining the homeostasis of different ecosystems", I believe there are more adequate literatures to cite, such as:
Ernest SKM and Brown JH. 2001. Homeostasis and Compensation: The Role of Species and Resources in Ecosystem Stability. Ecology, 82(8):2118-2132.
Thanks for the reference, it includes a very interesting approach to study the homeostasis in at ecosystem level . After reading we have reworded the paragraph.
In lines 48-54: the paragraph about the decline of honey bee colonies (and the different factors currently known and associated with CCD) should be better organized and expanded.
Details have been included
In item 2.1. Veterinary inspection: please write more clearly about the seasons and their corresponding months (beginning of autumn 2015, winter, early spring 2016).
Details included in the corresponding section of Material and Methods.
In item 2.2. Pathogen screening: it is really important to mention what was not detected in the samples (lines 109-112), but I recommend not putting the undetected parameters in the legend of Table 1.
The head of the table 1 has been updated .
In lines 118-119: "natural logarithm of the toxic unit of the mixture in each colony (Ln(TUm))" needs to be better explained somewhere in the manuscript.
Details included.
In item 2.1.1. Varroa mite resistance to acaricides (lines 122-126 and Table 2): the entire dataset needs to be clearly and better explained.
Details of the test are included in the Material and Method section. The results have been detailed.
The discussion section is interesting, but it could be more objective (summarized) and focused only on the case study data.
In this section we explained the possible cause of the collapse event ,based on the experimental results of screening analysis of pathogens, residue and palynological analysis of the beebread and climatic conditions, summarized in the section of Results In principle the beekeeper considered the main cause was the action of Varroa mites, which was ruled out due to the lack of symptomatic signs during the veterinary visit. Nevertheless, the infection levels of N.ceranae were very high which seem to be the main cause of the collapse of the apiary. Information on the diagnosis and consequences of the disease are included, which are in line with diagnosis signs observed during the visit.The evolution of the infection could be favoured by the presence of high levels of acaricides in the beebread and the unusual climatic conditions. The discussion includes some information on how these factors can favour the evolution of the disease.
The methodology is quite extensive but I still think it deserves a little more detail so that readers from other areas can understand the entire study.
Details of the methodology followed in the screening of pathogens and palynological identification of beebread are included in the section of Material and Methods. The chemical analysis is described in additional publications.
I recommend that authors add more collected data as supplementary material, such as PCR or RT-PCR images, pollen images, and so on.
We have included further details of the primers used in the pathogen screening and the pollen analysis in the supplementary information. We do not consider that including PCR images is necessary, since they are techniques normally used and that our working group has used in different papers already published, which demonstrates the reliability of our laboratory.
Reviewer 3 Report
The manuscript is very interesting, a huge effort was done for data collection influencing colony stress. Results are presented without detailed interpretation, analyses of the cumulative influence of each cause on colony losses (interaction) are needed.
Environmental data has a strong influence, it would be helpful to present it in the MS in order to better understand process and interaction.
Material and methods need to be written more clearly, it is not clear the number of samples for different analyses and the structure of samples.
The title of the tables needs to reflect data presented in the table. Some data mentioned in the title missing.
Author Response
Answer to Reviewer 3: As recommended by the referee environmental data have been included in the ms to better understand the interaction processes. The results showed the year of the collapse (2015) was characterized to be drier than the historic data, especially during the autumn winter seasons, and it had a wider amplitude temperature between the coldest and hottest months. This fact may have favoured to the infection by Nosema ceranae together with the accumulation of acaricides in the bee bread. The accumulative influence of these factors is commented the discussion section. Moreover, further details are included in the Material and Method section regarding to pathogen screening detection.
Annex: detailed answer to each comment.
The manuscript is very interesting, a huge effort was done for data collection influencing colony stress.
No action required
Results are presented without detailed interpretation, analyses of the cumulative influence of each cause on colony losses (interaction) are needed.
The interpretation is included in the Discussion section lines 240-253
Environmental data has a strong influence, it would be helpful to present it in the MS in order to better understand process and interaction.
A Walter-Lieth diagram was developed with historical weather data obtained from meteorological station of Brihuega (lat. 40.765, long.-2.874) included in the network of the The State Meteorological Agency (AEMET) to compare it with the data of the period of the study. This weather station is located at 9 km far away from the studied apiary. Walter & Leith climate diagrams are brief summaries of average climatic variables and their time course. They illustrate precipitation and temperature changes throughout the year in one standardized chart
The results showed the year of the collapse (2015) was characterized to be drier than the historic data, especially during the autumn winter seasons, and it had an wider amplitude temperature between the coldest and hottest months. This fact may have favoured to the infection by Nosema ceranae
The unsual climatic conditions of the year of the collapse event may have an in-fluence on the strength of the colonies. On one side, the warmer and drier conditions of this year may have provoked a change in the phenology and physiology of the vegeta-tion of the zone, with a decrease of the length of the flowering period the biodiversity and quality of the collected pollen and indirectly affecting on the strength of the colonies, favoring the infection by N. ceranae, which spores resist to high temperatures and desiccation and they complete its life cycle more efficiently at high temperatures. In these conditions, in addition, the multiplication of the parasite is favoured and the growth of the bee colony is hindered, which prevents the natural dilution of the disease, as indicated in Higes et al., 2008.
The discussion section has been updated, accordingly.
Material and methods need to be written more clearly, it is not clear the number of samples for different analyses and the structure of samples.
More details are included in the Material and Method section.
The title of the tables needs to reflect data presented in the table. Some data mentioned in the title missing.
The table was reduced in order to fit the outline of the editorial. The no detection of these pathogens is already mentioned in the text. The inclusion of this information in the table is redundant. The head of the table 1 has been updated .
References
- Higes, M.; Martín-Hernández, R.; Botías, C.; Bailón, E.G.; González-Porto, A.V.; Barrios, L.; del Nozal, M.J.; Bernal, J.L.; Jiménez, J.J.; Palencia, P.G.; et al. How natural infection by Nosema ceranae causes honeybee colony collapse. Environmental Microbiology 2008, 10, 2659-2669, doi:https://doi.org/10.1111/j.1462-2920.2008.01687.x.
- Costa, C.; Tanner, G.; Lodesani, M.; Maistrello, L.; Neumann, P. Negative correlation between Nosema ceranae spore loads and deformed wing virus infection levels in adult honey bee workers. Journal of Invertebrate Pathology 2011, 108, 224-225, doi:https://doi.org/10.1016/j.jip.2011.08.012.
- Dussaubat, C.; Brunet, J.-L.; Higes, M.; Colbourne, J.K.; Lopez, J.; Choi, J.-H.; Martín-Hernández, R.; Botías, C.; Cousin, M.; McDonnell, C.; et al. Gut Pathology and Responses to the Microsporidium Nosema ceranae in the Honey Bee Apis mellifera. PLOS ONE 2012, 7, e37017, doi:10.1371/journal.pone.0037017.
- Hedtke, K.; Jensen, P.M.; Jensen, A.B.; Genersch, E. Evidence for emerging parasites and pathogens influencing outbreaks of stress-related diseases like chalkbrood. Journal of Invertebrate Pathology 2011, 108, 167-173, doi:https://doi.org/10.1016/j.jip.2011.08.006.
- Martin, S.J.; Hardy, J.; Villalobos, E.; Martín-Hernández, R.; Nikaido, S.; Higes, M. Do the honeybee pathogens Nosema ceranae and deformed wing virus act synergistically? Environmental Microbiology Reports 2013, 5, 506-510, doi:https://doi.org/10.1111/1758-2229.12052.
- Doublet, V.; Natsopoulou, M.E.; Zschiesche, L.; Paxton, R.J. Within-host competition among the honey bees pathogens Nosema ceranae and Deformed wing virus is asymmetric and to the disadvantage of the virus. Journal of Invertebrate Pathology 2015, 124, 31-34, doi:https://doi.org/10.1016/j.jip.2014.10.007.
- Bartolomé, C.; Buendía-Abad, M.; Benito, M.; Sobrino, B.; Amigo, J.; Carracedo, A.; Martín-Hernández, R.; Higes, M.; Maside, X. Longitudinal analysis on parasite diversity in honeybee colonies: new taxa, high frequency of mixed infections and seasonal patterns of variation. Scientific Reports 2020, 10, 10454, doi:10.1038/s41598-020-67183-3.
- Buendía-Abad, M.; Higes, M.; Martín-Hernández, R.; Barrios, L.; Meana, A.; Fernández Fernández, A.; Osuna, A.; De Pablos, L.M. Workflow of Lotmaria passim isolation: Experimental infection with a low-passage strain causes higher honeybee mortality rates than the PRA-403 reference strain. International Journal for Parasitology: Parasites and Wildlife 2021, 14, 68-74, doi:https://doi.org/10.1016/j.ijppaw.2020.12.003.
- Buendía-Abad, M.; García-Palencia, P.; Pablos-Torró, L.M.d.; Alunda, J.M.; Osuna, A.; Martín-Hernández, R.; Higes, M. First description of Lotmaria passim and Crithidia mellificae haptomonad stage in the honeybee hindgut. bioRxiv 2021.
- Alonso-Prados, E.; Muñoz, I.; De la Rúa, P.; Serrano, J.; Fernández-Alba, A.R.; García-Valcárcel, A.I.; Hernando, M.D.; Alonso, Á.; Alonso-Prados, J.L.; Bartolomé, C.; et al. The toxic unit approach as a risk indicator in honey bees surveillance programmes: A case of study in Apis mellifera iberiensis. Sci Total Environ 2020, 698, 134208, doi:10.1016/j.scitotenv.2019.134208.
- Gómez-Moracho, T.; Buendía-Abad, M.; Benito, M.; García-Palencia, P.; Barrios, L.; Bartolomé, C.; Maside, X.; Meana, A.; Jiménez-Antón, M.D.; Olías-Molero, A.I.; et al. Experimental evidence of harmful effects of Crithidia mellificae and Lotmaria passim on honey bees. International Journal for Parasitology 2020, 50, 1117-1124, doi:https://doi.org/10.1016/j.ijpara.2020.06.009.
- Cepero, A.; Ravoet, J.; Gómez-Moracho, T.; Bernal, J.L.; Del Nozal, M.J.; Bartolomé, C.; Maside, X.; Meana, A.; González-Porto, A.V.; de Graaf, D.C.; et al. Holistic screening of collapsing honey bee colonies in Spain: a case study. BMC Research Notes 2014, 7, 649, doi:10.1186/1756-0500-7-649.
- Stevanovic, J.; Schwarz, R.S.; Vejnovic, B.; Evans, J.D.; Irwin, R.E.; Glavinic, U.; Stanimirovic, Z. Species-specific diagnostics of Apis mellifera trypanosomatids: A nine-year survey (2007–2015) for trypanosomatids and microsporidians in Serbian honey bees. Journal of Invertebrate Pathology 2016, 139, 6-11, doi:https://doi.org/10.1016/j.jip.2016.07.001.
Round 2
Reviewer 1 Report
Dear Authors,
I am satisfied with the changes and responses to my comments/suggestions made in the previous version of this manuscript. Now the work looks much more complete.
Reviewer 3 Report
Data presented in the article is a good base for understanding colony losses in other regions and countries with similar climate conditions.